analytical chemistry/environmental chemistry/green chemistry

aerosol, PAHs, alternating trilinear decomposition-assisted multivariate curve resolution, multivariate curve resolution-alternating least-squares, GC-MS

**Authors for correspondence:**
Ling Xu
e-mail: xuling@hncu.edu.cn
Xiaohua Zhang
e-mail: xhzhang12016020@xcu.edu.cn

This article has been edited by the Royal Society of Chemistry, including the commissioning, peer review process and editorial aspects up to the point of acceptance.

# Application of alternating trilinear decomposition-assisted multivariate curve resolution to gas chromatography-mass spectrometric data for the quantification of polycyclic aromatic hydrocarbons in aerosols

Xiangdong Qing[1], Xiaohong Zhou[1], Ling Xu[1], Jin Zhang[1], Yi Huang[1], Li Lin[1], Zhi Liu[2] and Xiaohua Zhang[3]

[1]Hunan Provincial Key Laboratory of Dark Tea and Jin-hua, College of Materials and Chemical Engineering, Hunan City University, Yiyang, 413000, People's Republic of China
[2]College of Agriculture and Biotechnology, Hunan University of Humanities, Science and Technology, Loudi, 417000, People's Republic of China
[3]Key Laboratory of Biomarker Based Rapid-detection Technology for Food Safety of Henan Province, Food and Bioengineering College, Xuchang University, Xuchang, 461000, People's Republic of China

LX, 0000-0003-4879-7605

For the first time, alternating trilinear decomposition-assisted multivariate curve resolution (ATLD-MCR) was applied to analyse complex gas chromatography–mass spectrometric (GC-MS) data with severe baseline drifts, serious co-elution peaks and slight retention time shifts for the simultaneous identification and quantification of polycyclic aromatic hydrocarbons (PAHs) in aerosols. It was also compared with the classic multivariate curve resolution-alternating least-squares (MCR-ALS) and the GC-MS-based external standard method. In validation samples, average recoveries of five PAHs were within the range from $(96.2 \pm 6.8)\%$ to $(106.5 \pm 4.1)\%$ for ATLD-MCR, near to the results of MCR-ALS

((98.0 ± 1.5)% to (106.7 ± 4.3)%). In aerosol samples, the concentrations of pyrene provided by ATLD-MCR were not significantly different from those of MCR-ALS. The other four PAHs including chrysene, benzo[a]anthracene, fluoranthene and benzo[b]fluoranthene were not detected by ATLD-MCR and the GC-MS-based external standard method. The results of figures of merit further demonstrated that ATLD-MCR achieved high sensitivities ($8.9 \times 10^4$ to $1.7 \times 10^6$ mAU ml µg$^{-1}$) and low limits of detection (0.003 to 0.087 µg ml$^{-1}$), which were better than or similar to MCR-ALS, presenting a great choice to deal with complex GC-MS data for the simultaneous determination of targeted PAHs in aerosols.

# 1. Introduction

Polycyclic aromatic hydrocarbons (PAHs), which are known to have biotoxicity such as carcinogenicity, mutagenicity and teratogenicity to human and organisms, are extensively distributed in atmospheric aerosols [1–4]. In recent years, the accurate quantification of PAHs in atmospheric aerosols has become one of the hottest topics among atmospheric science. However, atmospheric aerosols contain many chemical components which have a very low concentration and seriously mutual interference, making the accurate determination of PAHs in actual systems hard or impossible even with the help of modern analytical instruments, such as gas chromatography–mass spectrometry (GC-MS).

GC-MS is widely applied in the identification and quantification of volatile organic pollution in actual samples [5–8]. It has advantages of high sensitivity and accurate qualitative and quantitative abilities. Nevertheless, these excellent capacities of GC-MS may be discounted in the analysis of complex natural matrices due to some disturbing factors, such as baseline drifts, retention time shifts, non-Gaussian peaks, peak shape variations, unexpectedly overlapped peaks and so on. To reduce or even eliminate these disturbing factors, a commonly used strategy is to carry out a tedious sample pretreatment, which includes the extraction, concentration, purification and complete separation of analytes of interest. However, this will make the researchers devote more time, money and experimental efforts to a study.

Thus, the interference-free detection methods, which can carry out the accurate determination of targeted analytes in actual samples even in the presence of unknown and uncalibrated interferences, need to be developed. Fortunately, the chemometric three-way calibration method gives us a great choice for developing this type of approaches, which has 'second-order advantage' [9–12]. Up to now, three-way calibration methods can be divided into three categories based on different decomposition principles. The first is built on the alternating least-squares principle for the trilinear decomposition of three data arrays, for example, the classical parallel factor analysis (PARAFAC) [13], alternating trilinear decomposition (ATLD) [14] and its different versions (self-weighted ATLD [15] and penalized ATLD [16]). The second is based on the direct least-squares principle and residual bilinearization (RBL) for an augmented matrix, including unfolded partial least-squares combined with RBL (U-PLS/RBL) [17], N-way partial least-squares/RBL (N-PLS/RBL) [18] and unfolded principal component analysis/RBL (U-PCA/RBL) [19]. The third is built on the alternating least-squares and bilinear decomposition of an augmented matrix, known as multivariate curve resolution coupled to alternating least-squares (MCR-ALS) [20], which allows for the multi-linearity deviation of three-way data. Additionally, the newly proposed method of alternating trilinear decomposition-multivariate curve resolution (ATLD-MCR) [21], which fully uses the advantages of ATLD and MCR-ALS, can be included in the last type of methods.

ATLD-MCR achieves great success in dealing with three-way liquid chromatographic data with retention time shifts [11,21,22]. However, no study so far has demonstrated that ATLD-MCR can process complex GC-MS data with severe baseline drifts, overlapped peaks and retention time shifts. In the present work, ATLD-MCR was applied to analyse complex GC-MS data for the first time. First, the validation set containing five PAHs at three concentration levels within the calibration range was used to investigate the performance of ATLD-MCR. Then, actual aerosol three-way data were analysed by ATLD-MCR for the simultaneous identification and quantification of PAHs in aerosol samples. Moreover, ATLD-MCR was compared with the classic algorithm such as MCR-ALS, and concentrations of PAHs in aerosol samples predicted by ATLD-MCR were confirmed by the GC-MS-based external standard method.

# 2. Material and methods

## 2.1. Reagents and chemicals

Standard substances of fluoranthene (FLO) and benzo[b]fluoranthene (BbF) were purchased from AccuStandard Inc. (Connecticut, USA). Standard substances of chrysene (CHR), benzo[a]anthracene (BaA) and pyrene (PYR) were obtained from Aladdin Chemical Co. Ltd (Shanghai, China). N-hexane (AR) and dichloromethane (AR) were purchased from Hui-hong Reagent Corporation (Changsha, China). Methanol (HPLC-grade, ≥99.9%) was provided by Aladdin.

The analytical standard of BbF was dissolved with a few millilitres of dichloromethane after being accurately weighed, then diluting with methanol in a 100 ml volumetric flask. Stock solutions of FLO, CHR, BaA and PYR were prepared by accurately weighing each corresponding analytical standard and directly dissolving into methanol and diluting with methanol in a 100 ml brown volumetric flask. These prepared solutions were kept in a refrigerator at 4.0°C until used.

## 2.2. Instrumentation

All samples were measured on the GC-MS (GC/MS-QP2010, Shimadzu Corporation, Japan) with an Rxi®-1MS (Restek, USA) non-polar capillary column (0.25 µm and 0.25 mm × 30 m) as the separation column. The temperature-rising program was used and set to initiate the temperature at 150°C and hold for 2.0 min, then increase at a rate of 30°C min$^{-1}$ to 300°C and hold for 3.0 min. The electron impact (EI) source was applied as the ion source of the mass spectrum detector under the scan mode ($m/z$ of 50–400 amu). Both the interface and injector temperatures were 250°C. The ion source temperature and the detection voltage were 230°C and 0.94 kV, respectively. The carrier gas was helium (He) at a rate of 1.0 ml min$^{-1}$.

## 2.3. Sample collection and preparation

Aerosol samples (particulate matter with dp < 10 µm, PM10) were collected with an air sampler (YH-1000, Jingcheng Corporation, China) from Loudi City, China, for 8 h at five functional zones including railway station, Lian-Gang steel-making plant, municipal government, Shu-Yi building and Bi-Wu building. Five PM10 samples were captured onto quartz-fibre filters (20 × 24 cm). For the determination of PAHs in aerosols, each filter sample was extracted by a dichloromethane (60%) and n-hexane (40%) mixture with the Soxhlet method. Each extract was concentrated by rotary evaporation (40°C) to 2.0 ml after being filtered.

## 2.4. Sample sets

A calibration set including seven calibration samples (C01–C07) was constructed by accurately pipetting suitable amounts of working solutions of five PAHs into 10.0 ml brown volumetric flasks and diluting to the mark with methanol. Three validation samples (V01–V03) were prepared by using the same method as calibration samples, but the final analyte concentrations in validation samples should be within the concentration range of calibration samples, which are summarized in table 1. The extracted solutions of aerosol samples were used to analyse the contents of PAHs by GC-MS. Additionally, the concentrations of each PAH, which were employed to build standard curves as a function of the GC peak height in the GC-MS-based external standard method, were the same as those of calibration samples (C01–C07) shown in table 1. All samples were filtered through to 0.22 µm non-sterile PTFE syringe filter before being injected into the GC-MS column.

## 2.5. Method and software

Theories of the applied multi-way calibration models are well documented [14,20,21]. The ATLD-MCR program was provided by Tong Wang [21]. MCR-ALS was performed on the free software of the MVC2 toolbox supplied by Alejandro C. Olivieri [23]. The recorded GC-MS data were analysed by applying the Matlab R2015b (The Math Works, Inc., Natick, MA, USA) software. All calculations were implemented with a Lenovo 1.80 GHz Intel® Core™ i5-8250 U CPU with 8.00 GB RAM under Microsoft Windows 10 operating system.

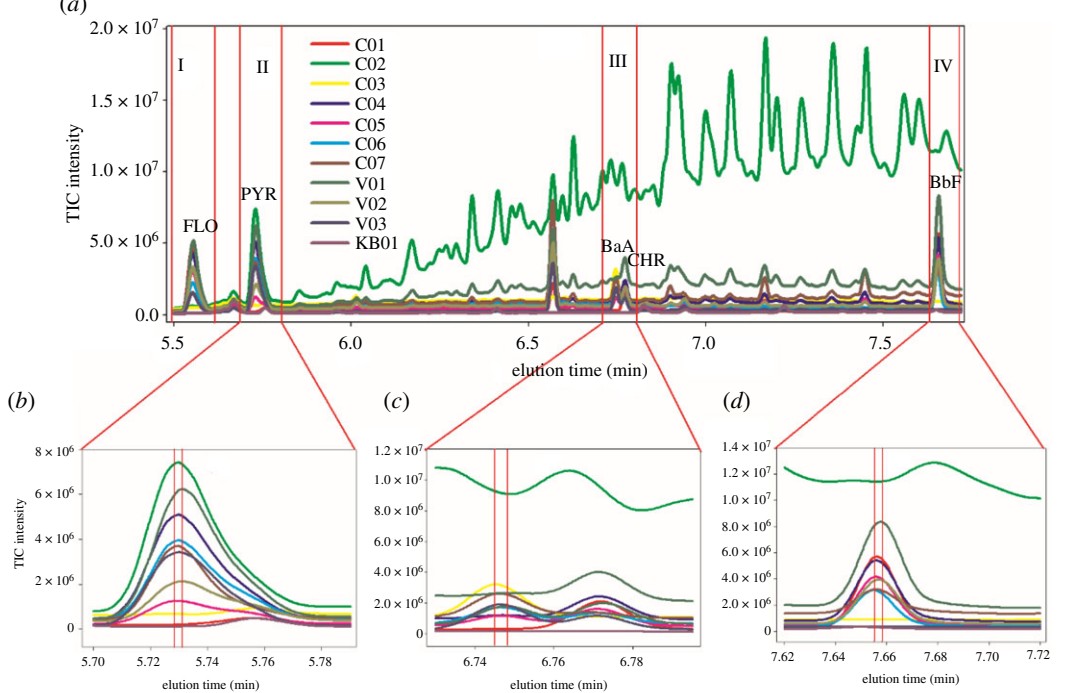

**Figure 1.** (*a*) Total ion current (TIC) chromatograms of seven calibration samples (C01–C07), three validation samples (V01–V03) and one blank sample (KB01); (*b*), (*c*) and (*d*) are the partially enlarged views of the II, III and IV regions, respectively.

**Table 1.** Concentrations of five PAHs in seven calibration (C01–C07) and three validation (V01–V03) samples.

| sample no. | analyte concentration ($\mu g\ ml^{-1}$) | | | | |
|---|---|---|---|---|---|
| | FLO | PYR | BaA | CHR | BbF |
| C01 | 5.60 | 0.00 | 0.00 | 1.00 | 3.42 |
| C02 | 0.00 | 6.40 | 0.00 | 0.00 | 0.00 |
| C03 | 0.00 | 0.00 | 0.68 | 0.00 | 0.00 |
| C04 | 4.48 | 4.99 | 0.14 | 0.80 | 2.85 |
| C05 | 3.36 | 1.24 | 0.27 | 0.60 | 2.28 |
| C06 | 2.24 | 3.84 | 0.41 | 0.40 | 1.71 |
| C07 | 1.12 | 2.56 | 0.54 | 0.20 | 1.14 |
| V01 | 5.04 | 5.76 | 0.20 | 0.90 | 3.99 |
| V02 | 3.36 | 1.92 | 0.41 | 0.70 | 2.00 |
| V03 | 1.68 | 3.84 | 0.61 | 0.50 | 0.00 |

# 3. Results and discussion

## 3.1. Gas chromatographic analysis

Figure 1 depicts the chromatographic profiles of seven calibration samples (C01–C07), three validation samples (V01–V03) and one blank sample (KB01). As could be seen from figure 1*a*, five targeted analytes, including FLO, PYR, BaA, CHR and BbF, were eluted in the time range of 5.50 to 7.72 min. In the validation samples, there were some co-elution phenomena for these compounds, e.g. the overlapped peaks between BaA and CHR. However, there was a more complex situation for actual PM10 samples, which are shown in figure 2. Due to other organic components in PM10, there were more co-eluted peaks than validation samples. For the complex system, it was very difficult or impossible to implement

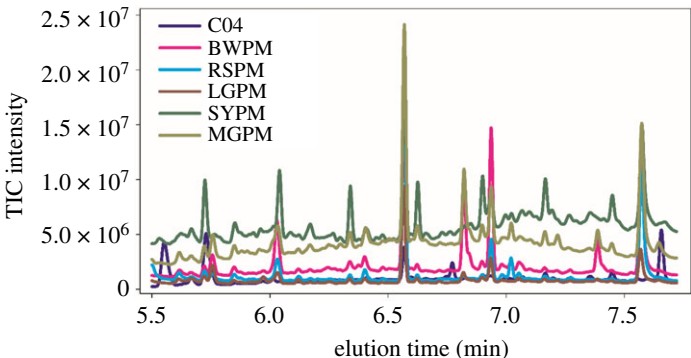

**Figure 2.** TIC chromatograms of five aerosol samples and the fourth calibration sample (C04). BWPM: PM10 in Bi-Wu building; RSPM: PM10 in railway station; LGPM, PM10 in Lian-Gang steel-making plant; SYPM: PM10 in Shu-Yi building; MGPM: PM10 in municipal government.

an accurate determination of analytes of interest applying general chromatographic methods and the univariate regression. Thus, it was worth exploiting the second-order advantage to satisfactorily solve the above-mentioned problems by applying the chemometric three-way calibration method.

Before decomposing the GC-MS data, the consideration of general characteristics was needed. In the GC analysis, baseline drifts and retention time shifts of chromatographic elution profiles were common to observe. As shown in figure 1a, the chromatographic baselines of most samples drifted seriously and differed from run to run in the studied elution domain. According to our experience, the baseline drift of GC-MS could be efficiently processed by the three-way calibration strategy, which was proved in our previous work [5]. Additionally, slight retention time shifts could be found because of analytes' retention time variations from run to run through the temperature program mode (figure 1b–d). ATLD-MCR was recently developed by Wang et al. [21] to process three-way liquid chromatographic data array with slight or even severe retention time shifts and achieved great success in resolving this type of problem. However, it was unknown whether ATLD-MCR could handle the complex GC-MS data with unknown overlapped peaks and retention time shifts or not, and how the performance of ATLD-MCR was in the analysis of this type of three-way data. Thus, it was worth further investigating the performance of ATLD-MCR with regard to processing GC-MS three-way data for the fast detection of PAHs even in the presence of unknown and uncalibrated interferences. Therefore, in the following sections, ATLD-MCR was employed to decompose the GC-MS three-way data array for the identification and quantification of five PAHs in validation and aerosol samples, and it was compared with the classic methods, such as MCR-ALS and the GC-MS-based external standard method.

## 3.2. Identification and quantification of polycyclic aromatic hydrocarbons in validation samples

To remove invalid information and provide accurate solutions, the retention region of five PAHs was divided into four sub-segments, including 5.50 to 5.62 min for FLO (I), 5.63 to 5.79 min for PYR (II), 6.72 to 6.79 min for BaA and CHR (III), and 7.62 to 7.72 min for BbF (IV), respectively, which are shown in figure 1a. Then, four three-way data arrays could be obtained. The core consistency diagnostic (CORCONDIA) test [24] was firstly employed to pre-estimate the number of chemical components (N). According to the provided results of CORCONDIA, the adopted N values in the decomposition of four three-way data arrays with the two algorithms all were three except that the III sub-segment was eight for MCR-ALS.

For the first and second regions (figure 1a, (I, II)), the identified and quantified results of FLO and PYR with ATLD-MCR as well as MCR-ALS are given in figure 3 (the resolved profiles of FLO and PYR are shown in one figure for the comparison of them) and table 2 (third and fourth columns). For FLO, ATLD-MCR gave good qualitative and quantitative results similar to MCR-ALS (see the blue solid lines in figure 3 and the third column of table 2). For PYR, although there were clear baseline drifts, unknown co-eluted interferences and slight retention time shifts (figure 1b), ATLD-MCR extracted clear chromatographic, mass spectral and relative concentration profiles of PYR in all samples (figure 3 (a1-2, b1 and c1)). And the retention profiles of PYR resolved by ATLD-MCR better reflected the variation of the chromatographic peak shape and the degree of the retention time shift of PYR in each sample (the red solid lines in figure 3 (a1-2)). These factors directly led to the deviation

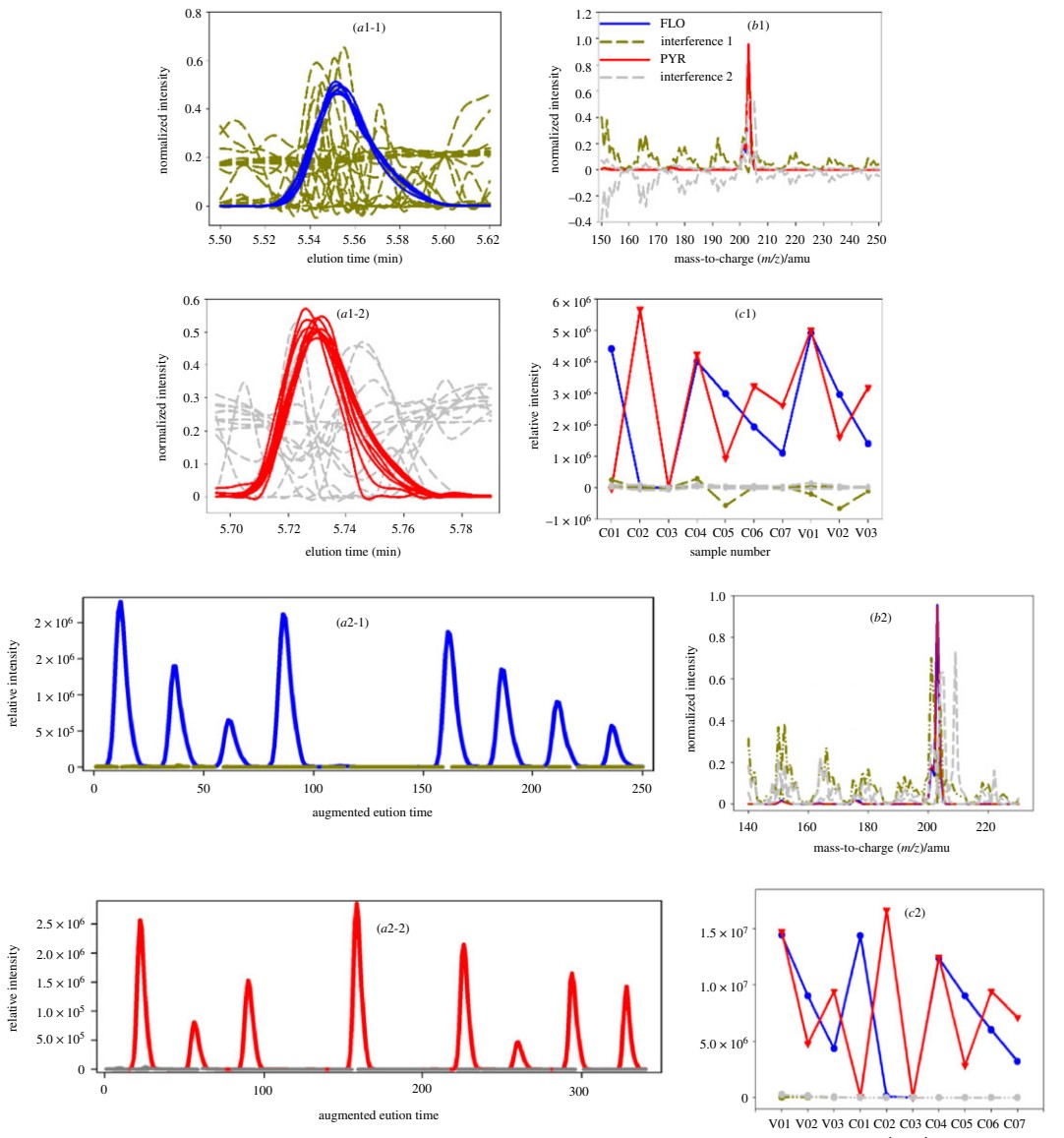

**Figure 3.** Chromatographic, spectral and relative concentration profiles of FLO and PYR in three validation samples retrieved by ATLD-MCR (a1, b1 and c1) and MCR-ALS (a2, b2 and c2), respectively.

of the trilinear structure of the sub-three-way data array. From table 2, it can be seen that the average recovery of PYR provided by ATLD-MCR was $(96.6 \pm 1.6)\%$ with RMSEP of $0.17\ \mu g\ ml^{-1}$, near to $(98.0 \pm 1.5)\%$ with RMSEP of $0.12\ \mu g\ ml^{-1}$ by MCR-ALS. The $t$-test values were 2.01 for ATLD-MCR and 1.33 for MCR-ALS, which indicated that the results of PYR resolved by ATLD-MCR and MCR-ALS were accurate and reliable.

For BaA and CHR (figure 1a,c), the third region (III)), not only were their chromatograms heavily overlapped with each other, but also there were severe baseline drifts and slight retention time shifts in them. However, ATLD-MCR overcame these problems and provided reasonable results. MCR-ALS also provided the resolved chromatographic, mass spectral and relative intensity profiles of them (figure 4 (a2, b2, c2)). But the results of ATLD-MCR better reflected the difference between each sample and the degree of the retention time shift in them (figure 4 (a1)). The quantitative and statistical results of ATLD-MCR were similar to those of MCR-ALS.

For BbF (the fourth region (IV)), there were slight retention time shifts between different samples (figure 1d), which were confirmed by the resolved chromatographic profiles of BbF by ATLD-MCR (figure 5 (a1)). And average recoveries were $(101.6 \pm 2.1)\%$ for ATLD-MCR, and $(103.5 \pm 1.5)\%$ for MCR-ALS. RMSEPs and $r^2$ were $0.15\ \mu g\ ml^{-1}$ and 0.9928 for ATLD-MCR, and $0.09\ \mu g\ ml^{-1}$ and 1.0000 for MCR-ALS, respectively.

**Table 2.** Quantitative and statistical results of two second-order calibration methods analysing five PAHs in validation samples.

| algorithm | sample | predicted concentration (µg mL$^{-1}$) (recovery (%)) | | | | |
|---|---|---|---|---|---|---|
| | | FLO | PYR | BaA | CHR | BbF |
| ATLD-MCR | V1 | 5.87 | 5.70 | 0.22 | 0.99 | 4.14 |
| | | [116.4] | [99.0] | [106.5] | [110.3] | [103.7] |
| | V2 | 3.47 | 1.86 | 0.38 | 0.76 | 1.99 |
| | | [103.2] | [96.7] | [92.6] | [109.0] | [99.6] |
| | V3 | 1.57 | 3.62 | 0.55 | 0.50 | — |
| | | [93.2] | [94.2] | [89.7] | [100.3] | — |
| | AR[a] | 104.2 | 96.6 | 96.2 | 106.5 | 101.6 |
| | AD[b] | 8.1 | 1.6 | 6.8 | 4.1 | 2.1 |
| | RMSEP[c] | 0.59 | 0.17 | 0.05 | 0.08 | 0.15 |
| | *t*-test[d] | 0.51 | 2.01 | 0.59 | 1.71 | 0.56 |
| | r$^{2e}$ | 0.9946 | 0.9966 | 0.9201 | 0.9707 | 0.9928 |
| MCR-ALS | V1 | 5.43 | 5.76 | 0.22 | 1.00 | 4.07 |
| | | [107.8] | [100.0] | [109.6] | [111.3] | [102.0] |
| | V2 | 3.39 | 1.89 | 0.43 | 0.76 | 2.10 |
| | | [100.8] | [98.3] | [105.5] | [108.6] | [105.0] |
| | V3 | 1.62 | 3.68 | 0.61 | 0.50 | — |
| | | [96.2] | [95.8] | [100.1] | [100.2] | — |
| | AR | 101.6 | 98.0 | 105.1 | 106.7 | 103.5 |
| | AD | 4.1 | 1.5 | 3.3 | 4.3 | 1.5 |
| | RMSEP | 0.28 | 0.12 | 0.02 | 0.08 | 0.09 |
| | *t*-test | 0.39 | 1.33 | 1.51 | 1.64 | 1.67 |
| | r$^2$ | 0.9988 | 0.9984 | 0.9890 | 0.9957 | 1.0000 |

[a]AR, average recovery (%).
[b]AD, average deviation.
[c]The root mean square error of prediction (RMSEP, µg ml$^{-1}$) can be calculated as follows:
$\mathrm{RMSEP} = \left[ 1/(I-1) \sum (c_n - \hat{c})^2 \right]^{1/2}$, where $c_n$ and $\hat{c}$ are the actual and predicted concentration in three validation samples, respectively, and $I$ is the number of samples.
[d]$T = (\bar{X} - \mu_0)/(S/\sqrt{n})$; here, $\bar{X}$ is the average recovery; $\mu_0$ is 1.0000, $n$ is the degree of freedom (where $n + 1$ is the number of evaluated levels), and the confidence level is 95%, $T_{0.025}^2 = 4.30$.
[e]$r^2$, correlation coefficient.

Compared with all the results provided by the two methods, it was found that ATLD-MCR and MCR-ALS could process complex GC-MS data and give reliable qualitative and quantitative results, and the results of ATLD-MCR were similar to those of MCR-ALS. Thus, in the following section, ATLD-MCR and MCR-ALS would be applied to identify and quantify PAHs in complex aerosol samples even in the presence of unknown and uncalibrated interferences.

## 3.3. Identification and quantification of five polycyclic aromatic hydrocarbons in aerosol samples

To assess the proposed methodology, five different types of PM10 samples collected from the local atmosphere were selected as real matrices. Concentrations of five PAHs in each aerosol sample were first detected by the GC-MS-based external standard method as the reference method. It was found that four PAHs including FLO, BaA, CHR and BbF were not detected by the reference method. Only PYR could be detected, and the predicted concentrations of PYR were 0.12, 0.13, 0.08, 0.12 and 0.06 µg ml$^{-1}$ for BWPM, RSPM, LGPM, MGPM and SYPM, respectively.

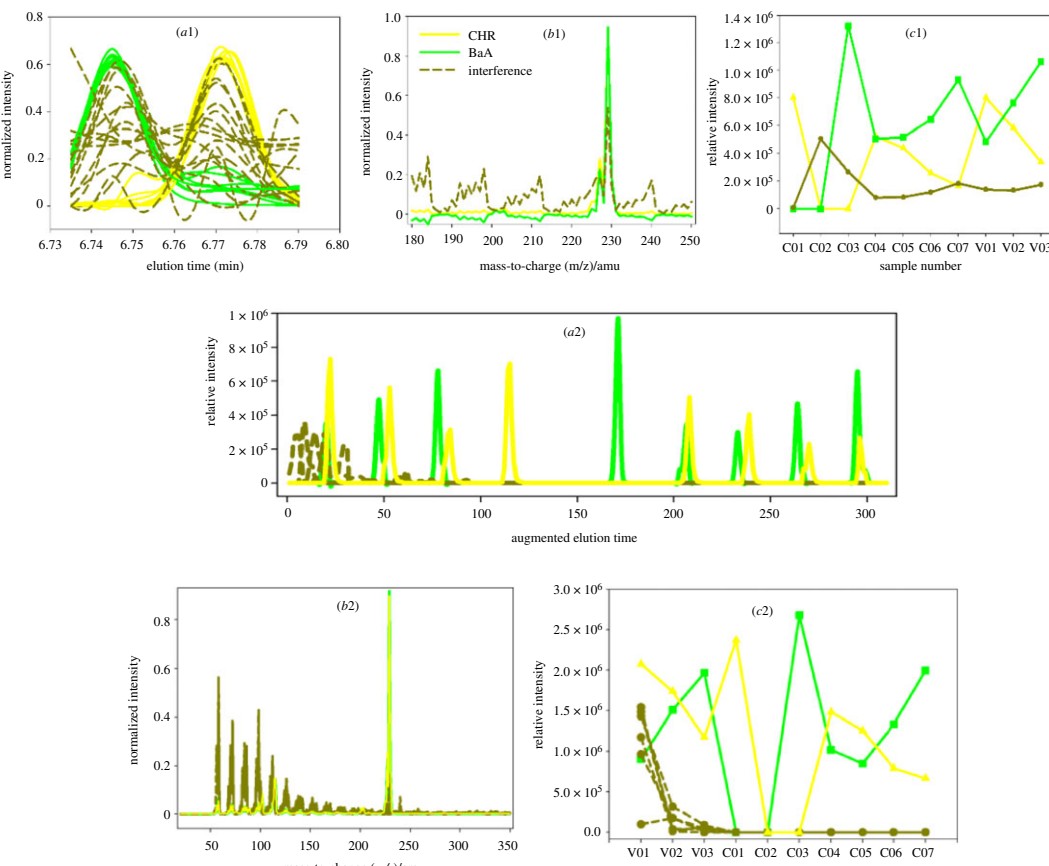

**Figure 4.** Chromatographic, spectral and relative concentration profiles of CHR and BaA in three validation samples retrieved by ATLD-MCR (*a*1, *b*1 and *c*1) and MCR-ALS (*a*2, *b*2 and *c*2), respectively.

ATLD-MCR and MCR-ALS were applied to construct calibration models for the identification and quantification of five PAHs in aerosols. A three-way dataset was firstly formed by stacking the elution time-mass spectral matrices for seven calibration samples and five aerosol samples. Then, four sub-three-way data arrays could be obtained by dividing the big three-way data array according to four elution segments above-mentioned. The CORCONDIA test was employed to determine the underlying number of chemical components in the actual system. For two algorithms, the factor numbers of four sub-three-way data arrays were 4, 4, 8 and 4, respectively. In the analysis of aerosol samples with the two algorithms, there were more unknown and uncalibrated interferences in the studied domain, seen from figure 2. However, thanks to the 'second-order advantage', the two algorithms could provide an accurate and reliable resolution of gas chromatograms and mass spectrograms of five PAHs in aerosol samples, which were similar to those in validation samples (not shown here).

Based on the reliable qualitative results in the chromatogram and mass spectrogram of each analyte, the quantification of five PAHs in aerosol samples could be provided by the two regression methods. For ATLD-MCR, the PYR concentrations in five aerosol samples were 0.16, 0.16, 0.10, 0.13 and 0.10 µg ml$^{-1}$, respectively; the predicted values of the other four PAHs were lower than the values of the limit of detection (LOD) of the corresponding PAH, which were regarded as 'no detection'. MCR-ALS, FLO, BaA and CHR were not detected in all aerosol samples; BbF were detected in LGPM and MGPM for predicted values of 0.008 and 0.011 µg ml$^{-1}$, respectively; the PYR concentrations in five aerosol samples were 0.16, 0.18, 0.10, 0.13 and 0.10 µg ml$^{-1}$, respectively, which were very near to those of ATLD-MCR. These results indicated that the results of ATLD-MCR were not significantly different from those of MCR-ALS. Also, all the predicted concentrations of five PAHs in aerosols by the two methods were lower than the values of the limit of quantification (LOQ) of the corresponding analyte. These results demonstrated that the contents of the targeted PAHs were very low in actual aerosol samples.

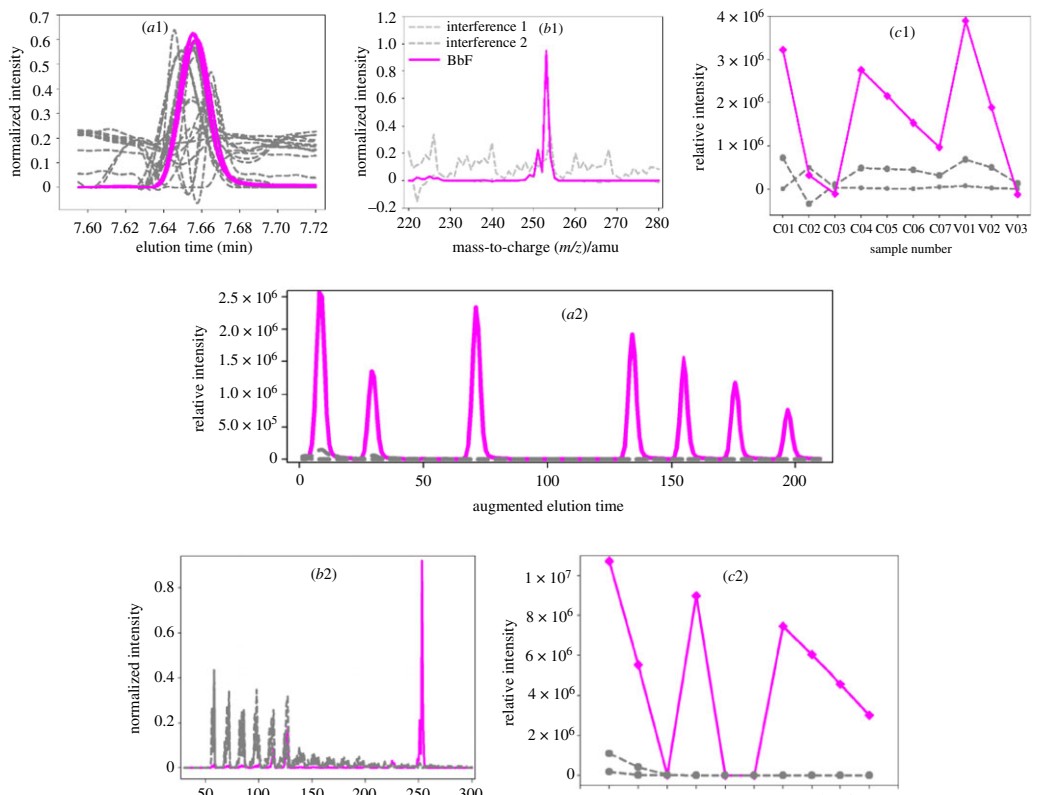

**Figure 5.** Chromatographic, spectral and relative concentration profiles of BbF in three validation samples retrieved by ATLD-MCR (*a*1, *b*1 and *c*1) and MCR-ALS (*a*2, *b*2 and *c*2), respectively.

It is worth noting that BaA and CHR severely overlapped with each other, and they could not be completely separated by the chromatographic procedure in the very long run time. In comparing ATLD-MCR with the reference method, the experimental time (10 min per sample) of ATLD-MCR was significantly shorter than the one of the reference method (more than 60 min per sample). Also, ATLD-MCR took fewer organic solvents and cost less than the reference method because the ATLD-MCR method did not need a tedious purification and complete separation of analytes of interest.

## 3.4. Compared with figures of merit of two algorithms

Then, some figures of merit were applied to evaluate and compare the performances of ATLD-MCR and MCR-ALS [25,26]. For the determination of FLO, PYR, BaA and CHR, ATLD-MCR provided higher sensitivities (SENs) than MCR-ALS, which were $6.25 \times 10^5$, $16.62 \times 10^5$, $1.90 \times 10^5$ and $0.89 \times 10^5$ mAU ml $\mu g^{-1}$ for ATLD-MCR, and $2.00 \times 10^5$, $2.10 \times 10^5$, $0.23 \times 10^5$ and $0.15 \times 10^5$ mAU ml $\mu g^{-1}$ for MCR-ALS, respectively. For BbF, the obtained SEN of $2.64 \times 10^5$ mAU ml $\mu g^{-1}$ by ATLD-MCR was near to the one of MCR-ALS ($3.40 \times 10^5$ mAU ml $\mu g^{-1}$). Two algorithms significantly improved the selectivities (SELs) of these analytes, especially for BaA and CHR, whose SELs were only 0.011 and 0.0094 for ATLD-MCR and 0.053 and 0.052 for MCR-ALS, respectively. Moreover, low LOD and LOQ values were obtained with two algorithms. It was found that the results of LOD and LOQ of ATLD-MCR were better than those of MCR-ALS except for BbF (the fourth to fifth, and eighth to ninth rows of table 3). The LOD of BbF was 0.008 $\mu g$ ml$^{-1}$ for MCR-ALS, better than the one of 0.087 $\mu g$ ml$^{-1}$ for ATLD-MCR, that was the reason MCR-ALS could carry out the determination of BbF in actual aerosol samples but ATLD-MCR could not.

Additionally, the obtained LODs and LOQs by ATLD-MCR were compared with those of two national standard methods, such as GC-MS [27] and high-performance liquid chromatography–ultraviolet visible (HPLC-UV) [28] (the last four rows of table 3). It was found that the LODs and LOQs provided by ATLD-MCR were lower than or close to those of GC-MS and HPLC-UV. However, the achieved results could be considered satisfactory for the simultaneous determination of five PAHs

**Table 3.** Figures of merit of two second-order calibration algorithms as well as two national standard methods in the analysis of five PAHs.

| method | figures of merit | FLO | PYR | BaA | CHR | BbF |
|---|---|---|---|---|---|---|
| ATLD-MCR | SEN[a] | 62.5 | 166.2 | 19.0 | 8.9 | 26.4 |
| | SEL[b] | 0.077 | 0.19 | 0.011 | 0.0094 | 0.028 |
| | LOD[c] | 0.017 | 0.020 | 0.003 | 0.007 | 0.087 |
| | LOQ[c] | 0.052 | 0.061 | 0.008 | 0.020 | 0.264 |
| MCR-ALS | SEN | 20.0 | 21.0 | 2.3 | 1.5 | 34.0 |
| | SEL | 0.99 | 0.97 | 0.053 | 0.052 | 0.99 |
| | LOD | 0.023 | 0.061 | 0.010 | 0.012 | 0.008 |
| | LOQ | 0.071 | 0.184 | 0.029 | 0.038 | 0.023 |
| GC-MS | LOD | 0.050 | 0.050 | 0.100 | 0.070 | 0.120 |
| | LOQ | 0.200 | 0.200 | 0.400 | 0.028 | 0.480 |
| HPLC-UV | LOD | 0.020 | 0.015 | 0.017 | 0.015 | 0.020 |
| | LOQ | 0.080 | 0.060 | 0.070 | 0.060 | 0.080 |

[a]The sensitivity (SEN, $10^4$ mAU ml $\mu g^{-1}$) is determined as: $SEN_{F03} = s_n \left\{ \left[ (A_{cal}^T P_{A,unx} A_{cal}) * (B_{cal}^T P_{B,unx} B_{cal}) \right]^{-1} \right\}_{nn}^{-1/2}$, where nn designates the (n,n) element of a matrix; $s_n$ is the total signal for component '$n$' at unit concentration.
[b]The selectivity (SEL) can be simply obtained by dividing SEN by $s_n$.
[c]The LOD ($\mu g$ $ml^{-1}$) and the LOQ ($\mu g$ $ml^{-1}$) are calculated as LOD = 3.3S(0), LOQ = 10S(0), respectively, where S(0) is the standard deviation in the predicted concentration of the analyte of interest in three blank samples.

in complex systems if baseline drifts, retention time shifts, peak shape variations and co-elution peaks of the complex GC-MS analysis system was taken into account.

# 4. Conclusion

In GC-MS analysis of complex aerosol samples, many disturbing factors such as baseline drifts, retention time shifts, non-Gaussian peaks, peak shape variations and overlapped peaks often make an accurate qualitative and quantitative determination of PAHs even more difficult or impossible. In the study, ATLD-MCR was applied to analyse complex GC-MS data with severe baseline drifts, serious co-elution peaks and slight retention time shifts for the first time, and it was compared with two classic methods, MCR-ALS and the GC-MS-based external standard method (the reference method). It was found in the study of validation samples that ATLD-MCR could give accurate and reliable results, which was similar to MCR-ALS. In the analysis of actual aerosol samples, the predicted results of PAHs except for BbF by ATLD-MCR were consistent with the ones of MCR-ALS and the reference method. These results of figures of merit further indicated that ATLD-MCR could provide high SENs, good SELs and low LODs and LOQs, which were better than or similar to MCR-ALS. The satisfactory results demonstrated that ATLD-MCR had advantages of directly decomposing three-way data array, the ability of dealing with non-trilinear data, providing pure chromatographic, spectral and relative concentration profiles of each chemical constituent of interest, second-order advantage and so on. It was an excellent choice for processing three-way GC-MS data with baseline drifts, overlapped peaks, peak shape variations and retention time shifts for the simultaneous real-time determination of the targeted PAHs in actual aerosol samples.

Data accessibility. The GC-MS three-way data files are provided in the Dryad Digital Repository at: https://doi.org/10.5061/dryad.kkwh70s47 [29].
Authors' contributions. X.Q. was involved in conceptualization, methodology, data curation,writing—original draft, review and editing. L.X. was involved in conceptualization, data curation and funding acquisition. X.Z. was involved in resources and validation. J.Z. was involved in software and validation. Y.H. was involved in methodology and validation. L.L. was involved in investigation. Z.L. was involved in methodology and software. X.Z. was involved in writing—review and editing and supervision.
Competing interests. The authors confirm that this article content has no conflicts of interest.

Funding. The authors gratefully acknowledge the National Natural Science Foundation of China (grant nos. 21707032 and 31701689), the China Scholarship Council (grant no. 202008430147), Hunan Provincial Natural Science Foundation (grant no. 2021JJ50151) and the Research Foundation of Education Bureau of Hunan Province, China (grant nos. 16A109 and 19C0345) for financial support.

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
