## [Peer Review File · Royal Society Open Science]

Review History

RSOS-210458.R0 (Original submission)

Review form: Reviewer 1

Is the manuscript scientifically sound in its present form?

Yes

Are the interpretations and conclusions justified by the results?

Yes

Is the language acceptable?

Yes

Do you have any ethical concerns with this paper?

No

Have you any concerns about statistical analyses in this paper?

No

Recommendation?

Accept with minor revision (please list in comments)

Comments to the Author(s)

Comments to Authors:

I have carefully reviewed the manuscript (Title: Application of alternating trilinear decomposition-assisted multivariate curve resolution to GC-MS data for the quantification of PAHs in aerosols). In this article, the author proposed a novel application of the alternating trilinear decomposition-assisted multivariate curve resolution (ATLD-MCR) for the analysis of GC-MS data to identify and quantify five PAHs in aerosols for the first time. Then, they also compared the performances of ATLD-MCR with two classic algorithms such as alternating trilinear decomposition (ATLD) and multivariate curve resolution couple to alternating least squares (MCR-ALS). It is a meaningful work, and current results would provide a powerful reference to deal with GC-MS data with baseline drifts, retention-time shifts and unexpected peaks overlapping using ATLD-MCR. This paper is suitable to publish in this journal after minor revision.

(1) In page 5 lines 38-40, the sentence "because there are many disturbing factors such as..." should be changed to "due to some disturbing factors, such as...". In page 7 lines 24-25, "were purchased by" should be modified to "were purchased from". In page 9 lines 33-35, "seven calibration (C01-C07), three validation (V01-V03)" should be changed to "seven calibration samples (C01-C07), three validation samples (V01-V03)". The author should further check and modify similar problem in the manuscript.

(2) In page 5 lines 48-49, "a tedious sample pretreatment" is not clear. It should be explained in more detail.

(3) In the "Introduction", the author state that "three-way calibration methods can be divided into three categories" (lines 7-9 in page 6), there are no examples for each type of method, so which algorithms to include in each type of method should be further elucidated for readers' understanding.

(4) Why does the retention region of five PAHs been divided into four sub-segments, but not three or two (Page 11 lines 17-20)?

(5) This manuscript should be submitted after careful inspection, such as duplicate line numbers.

(6) Although this manuscript was well-written, there are still some grammatical errors in the manuscript. For example, in page 16 lines 40-41, "...is taken into account" should be "...was taken into account". Therefore, language of this manuscript should be further polished.

(7) The format of references should be modified to meet the requirements of this journal.

Review form: Reviewer 2**Is the manuscript scientifically sound in its present form?**

Yes

Are the interpretations and conclusions justified by the results?

No

Is the language acceptable?

No

Do you have any ethical concerns with this paper?

No

Have you any concerns about statistical analyses in this paper?

No

Recommendation?

Major revision is needed (please make suggestions in comments)

Comments to the Author(s)

This manuscript described the application of gas chromatographic-mass spectral data to the analysis of polycyclic aromatic hydrocarbons in aerosol samples. Although the chemometric models are already known, the analytical application appears to be original, however the paper needs major revision.

- Only three validation samples (Table 1) are not enough to gather reliable statistical parameters and conclusions based on t-statistics. You may need to significantly increase the number of validation samples. I suggest a minimum of 10 samples.

- A GC-MS external reference method is mentioned several times in the text, but no details are provided for this technique. A comparison of the reference method and the one employed for three-way analysis is required, in terms of time, cost, equipment, etc.

- I do not really see the point in comparing ATLD, ATLD-MCR and MCR-ALS. ATLD is a trilinear decomposition technique which is known to be unreliable for non-trilinear data such as those presently studied. Ultimately, the analytical results from ATLD-MCR do not seem to be significantly better than those provided by MCR-ALS, which is the standard method for second-order chromatographic analysis of complex samples. I suggest removal of ATLD and ATLD-MCR from the text.

- In the real samples, Table 3, most of the analytes are undetected. Moreover, those that are detected appear to show concentrations close to the limit of detection (which in any case is an approximation, because the effects of slope and intercept have not been taken into account in estimating the LOD), and certainly below the limit of quantitation, which is not reported here. I guess most (if not all) concentration values quoted in Table 3 should be reported as "detected but not quantitated" because they are between the LOD and the LOQ. If this is the case, Table 3 is not required, and you can quote the results directly in the text. In a way, it is rather disappointing that no aerosol sample has been collected containing measurable amounts of the studied PAHs.

- In Figures 3, 4 and 5, some of the ATLD and ATLD-MCR profiles show negative elements, which is not acceptable for signals and concentrations which are all non-negative. Only MCR gives physically interpretable results, derived from the wise application of the non-negativity constraint. One more reason to remove the ATLD and ATLD-MCR results and the corresponding discussion on the theoretical and practical aspects of these two latter algorithms.

- The manuscript needs professional edition of the English.

Decision letter (RSOS-210458.R0)

Dear Ms Xu:

Title: Application of alternating trilinear decomposition-assisted multivariate curve resolution to GC-MS data for the quantification of PAHs in aerosols
Manuscript ID: RSOS-210458

The editor assigned to your manuscript has now received comments from reviewers. We would like you to revise your paper in accordance with the referee and Subject Editor suggestions which can be found below (not including confidential reports to the Editor). Please note this decision does not guarantee eventual acceptance.

Please submit your revised paper before 13-May-2021. Please note that the revision deadline will expire at 00.00am on this date. If we do not hear from you within this time then it will be assumed that the paper has been withdrawn. In exceptional circumstances, extensions may be possible if agreed with the Editorial Office in advance. We do not allow multiple rounds of revision so we urge you to make every effort to fully address all of the comments at this stage. If deemed necessary by the Editors, your manuscript will be sent back to one or more of the original reviewers for assessment. If the original reviewers are not available we may invite new reviewers.

Royal Society of Chemistry
Thomas Graham House
Science Park, Milton Road
Cambridge, CB4 0WF

Royal Society Open Science - Chemistry Editorial Office

RSC Associate Editor:
Comments to the Author:
(There are no comments.)

RSC Subject Editor:
Comments to the Author:
(There are no comments.)

Reviewers' Comments to Author:
Reviewer: 1

Comments to the Author(s)
Comments to Authors:

I have carefully reviewed the manuscript (Title: Application of alternating trilinear decomposition-assisted multivariate curve resolution to GC-MS data for the quantification of PAHs in aerosols). In this article, the author proposed a novel application of the alternating trilinear decomposition-assisted multivariate curve resolution (ATLD-MCR) for the analysis of GC-MS data to identify and quantify five PAHs in aerosols for the first time. Then, they also compared the performances of ATLD-MCR with two classic algorithms such as alternating trilinear decomposition (ATLD) and multivariate curve resolution couple to alternating least squares (MCR-ALS). It is a meaningful work, and current results would provide a powerful reference to deal with GC-MS data with baseline drifts, retention-time shifts and unexpected peaks overlapping using ATLD-MCR. This paper is suitable to publish in this journal after minor revision.

(1) In page 5 lines 38-40, the sentence "because there are many disturbing factors such as..." should be changed to "due to some disturbing factors, such as...". In page 7 lines 24-25, "were purchased by" should be modified to "were purchased from". In page 9 lines 33-35, "seven calibration (C01-C07), three validation (V01-V03)" should be changed to "seven calibration samples (C01-C07), three validation samples (V01-V03)". The author should further check and modify similar problem in the manuscript.

(2) In page 5 lines 48-49, "a tedious sample pretreatment" is not clear. It should be explained in more detail.

(3) In the "Introduction", the author state that "three-way calibration methods can be divided into three categories" (lines 7-9 in page 6), there are no examples for each type of method, so which algorithms to include in each type of method should be further elucidated for readers' understanding.

(4) Why does the retention region of five PAHs been divided into four sub-segments, but not three or two (Page 11 lines 17-20)?

(5) This manuscript should be submitted after careful inspection, such as duplicate line numbers.

(6) Although this manuscript was well-written, there are still some grammatical errors in the manuscript. For example, in page 16 lines 40-41, "...is taken into account" should be "...was taken into account". Therefore, language of this manuscript should be further polished.

(7) The format of references should be modified to meet the requirements of this journal.

Reviewer: 2

Comments to the Author(s)

This manuscript described the application of gas chromatographic-mass spectral data to the analysis of polycyclic aromatic hydrocarbons in aerosol samples. Although the chemometric models are already known, the analytical application appears to be original, however the paper needs major revision.

- Only three validation samples (Table 1) are not enough to gather reliable statistical parameters and conclusions based on t-statistics. You may need to significantly increase the number of validation samples. I suggest a minimum of 10 samples.

- A GC-MS external reference method is mentioned several times in the text, but no details are provided for this technique. A comparison of the reference method and the one employed for three-way analysis is required, in terms of time, cost, equipment, etc.

- I do not really see the point in comparing ATLD, ATLD-MCR and MCR-ALS. ATLD is a trilinear decomposition technique which is known to be unreliable for non-trilinear data such as those presently studied. Ultimately, the analytical results from ATLD-MCR do not seem to be significantly better than those provided by MCR-ALS, which is the standard method for second-order chromatographic analysis of complex samples. I suggest removal of ATLD and ATLD-MCR from the text.

- In the real samples, Table 3, most of the analytes are undetected. Moreover, those that are detected appear to show concentrations close to the limit of detection (which in any case is an approximation, because the effects of slope and intercept have not been taken into account in estimating the LOD), and certainly below the limit of quantitation, which is not reported here. I guess most (if not all) concentration values quoted in Table 3 should be reported as "detected but not quantitated" because they are between the LOD and the LOQ. If this is the case, Table 3 is not required, and you can quote the results directly in the text. In a way, it is rather disappointing that no aerosol sample has been collected containing measurable amounts of the studied PAHs.

- In Figures 3, 4 and 5, some of the ATLD and ATLD-MCR profiles show negative elements, which is not acceptable for signals and concentrations which are all non-negative. Only MCR gives physically interpretable results, derived from the wise application of the non-negativity constraint. One more reason to remove the ATLD and ATLD-MCR results and the corresponding discussion on the theoretical and practical aspects of these two latter algorithms.

- The manuscript needs professional edition of the English.

Author's Response to Decision Letter for (RSOS-210458.R0)

See Appendix A.

RSOS-210458.R1 (Revision)

Review form: Reviewer 1

Is the manuscript scientifically sound in its present form?

Yes

Are the interpretations and conclusions justified by the results?

Yes

Is the language acceptable?

Yes

Do you have any ethical concerns with this paper?

No

Have you any concerns about statistical analyses in this paper?

No

Recommendation?

Accept as is

Comments to the Author(s)

I have carefully reviewed the manuscript (Title: Application of alternating trilinear decomposition-assisted multivariate curve resolution to GC-MS data for the quantification of PAHs in aerosols, Manuscript ID: RSOS-210458.R1). According to the reviewer's suggestion, the manuscript has been revised in detail to make the article more rigorous. Hence, I recommend publishing this manuscript.

Review form: Reviewer 2

Is the manuscript scientifically sound in its present form?

Yes

Are the interpretations and conclusions justified by the results?

Yes

Is the language acceptable?

Yes

Do you have any ethical concerns with this paper?

No

Have you any concerns about statistical analyses in this paper?

No

Recommendation?

Accept as is

Comments to the Author(s)

The authors have responded to all issues raised by the reviewers. Although some specific points could not be addressed for different reasons, the authors have explained why they chose to accept the reviewers suggestions in some cases and why not in some others. Overall, I am satisfied with their responses.

Decision letter (RSOS-210458.R1)

Dear Ms Xu:

Title: Application of alternating trilinear decomposition-assisted multivariate curve resolution to GC-MS data for the quantification of PAHs in aerosols
Manuscript ID: RSOS-210458.R1

It is a pleasure to accept your manuscript in its current form for publication in Royal Society Open Science. The chemistry content of Royal Society Open Science is published in collaboration with the Royal Society of Chemistry.

RSC Associate Editor:
Comments to the Author:

(There are no comments.)

RSC Subject Editor:

Comments to the Author:

(There are no comments.)

Reviewer(s)' Comments to Author:

Reviewer: 2

Comments to the Author(s)

The authors have responded to all issues raised by the reviewers. Although some specific points could not be addressed for different reasons, the authors have explained why they chose to accept the reviewers suggestions in some cases and why not in some others. Overall, I am satisfied with their responses.

Reviewer: 1

Comments to the Author(s)

I have carefully reviewed the manuscript (Title: Application of alternating trilinear decomposition-assisted multivariate curve resolution to GC-MS data for the quantification of PAHs in aerosols, Manuscript ID: RSOS-210458.R1). According to the reviewer's suggestion, the manuscript has been revised in detail to make the article more rigorous. Hence, I recommend publishing this manuscript.

Appendix A

Ms Ling Xu

Hunan Provincial Key Laboratory of Dark Tea and
Jin-hua, College of Materials and Chemical Engineering,
Hunan City University, Yiyang, 413000, China

Fax: 86-737-6355075

Email: xuling@hncu.edu.cn

Dr Xiao-Hua Zhang

Key Laboratory of Biomarker Based Rapid- detection
Technology for Food Safety of Henan Province, Food and
Bioengineering College, Xuchang University, Xuchang,
461000, China

Fax:+863742968907

Email:xhzhang2329@126.com

May 5, 2021

Dr. Laura Smith

Journal Manager

Royal Society of Chemistry

E-mail: chemistryopenscience@rsc.org

Dear Editor,

Thank you very much for your valuable comments for the revision of our manuscript entitled “**Application of alternating trilinear decomposition-assisted multivariate curve resolution to GC-MS data for the quantification of PAHs in aerosols**” (Ms. No.: **RSOS-210458**).

According to the comments raised, we have made a careful revision of this manuscript, in which the changes are necessarily marked **in red**. All are responded in an item-by-item description in **Responses to Editor/Reviews**.

Best wishes,

Sincerely yours,

Ling Xu and Xiao-Hua Zhang (corresponding author)

Responses to Editor/Reviewers

Dear Editor/Reviewers,

Thank you very much for your valuable comments and suggestions for the revision of our manuscript to *Royal Society Open Science* (Ms. No.: *RSOS-210458*).

According to the comments raised, we have made a careful revision of this manuscript, in which the changes are necessarily marked **in red**. All are responded in a point-by-point description as follows.

Reviewer(s)' Comments:

Reviewer #1:

I have carefully reviewed the manuscript (Title: Application of alternating trilinear decomposition-assisted multivariate curve resolution to GC-MS data for the quantification of PAHs in aerosols). In this article, the author proposed a novel application of the alternating trilinear decomposition-assisted multivariate curve resolution (ATLD-MCR) for the analysis of GC-MS data to identify and quantify five PAHs in aerosols for the first time. Then, they also compared the performances of ATLD-MCR with two classic algorithms such as alternating trilinear decomposition (ATLD) and multivariate curve resolution couple to alternating least squares (MCR-ALS). It is a meaningful work, and current results would provide a powerful reference to deal with GC-MS data with baseline drifts, retention-time shifts and unexpected peaks overlapping using ATLD-MCR. This paper is suitable to publish in this journal after minor revision.

Authors' Response:

We do appreciate your kind approval on our work. According to your valuable comments and suggestions, we have checked and revised the whole manuscript carefully. The responses to your comments and suggestions are summarized as follows point-by-point. We hope that you will be satisfied with our responses and modifications.

(1) In page 5 lines 38-40, the sentence “because there are many disturbing factors such as...” should be changed to “due to some disturbing factors, such as...”. In page

7 lines 24-25, “were purchased by” should be modified to “were purchased from”. In page 9 lines 33-35, “seven calibration (C01-C07), three validation (V01-V03)” should be changed to “seven calibration samples (C01-C07), three validation samples (V01-V03)”. The author should further check and modify similar problem in the manuscript.

Authors’ Response:

Thank you very much for so careful and valuable comments. In the revised manuscript, we modified these sentences according to your suggestions. Please see the red words in page 4 lines 38-40, the sentence “because there are many disturbing factors such as...” was modified to “due to some disturbing factors, such as...”. In page 6 lines 40-41, “were purchased by” was modified to “were purchased from”. In page 8 lines 56-59, “seven calibration (C01-C07), three validation (V01-V03) and one blank (KB01) samples” was changed to “seven calibration samples (C01-C07), three validation samples (V01-V03) and one blank samples (KB01)”. Additionally, we think that writing quality of a scientific paper is very important for its spreading and influence. So, we further checked and modified the similar problem in the revised manuscript.

(2) In page 5 lines 48-49, “a tedious sample pretreatment” is not clear. It should be explained in more detail.

Authors’ Response:

Thanks for your suggestion. In the revised manuscript, we added one sentence (“which includes the extraction, concentration, purification and complete separation of analytes”) to further explain the phrase “a tedious sample pretreatment”. Please see the red words in page 4 lines 48-51.

(3) In the “Introduction”, the author state that “three-way calibration methods can be divided into three categories” (lines 7-9 in page 6), there are no examples for each type of method, so which algorithms to include in each type of method should be further elucidated for readers’ understanding.

Authors’ Response: Thank you for your valuable comments. According to your suggestion, we took a more description about three categories of three-way calibration

method. We added the following sentences into the revised manuscript: “The first is built on the alternating least-squares principle for the trilinear decomposition of three data arrays, for examples, the classical parallel factor analysis (PARAFAC), alternating trilinear decomposition (ATLD) and its different versions (self-weighted ATLD¹⁵ and penalized ATLD). The second is based on the direct least-squares principle and residual bilinearization (RBL) for an augmented matrix, including unfolded partial least-squares combined with RBL (U-PLS/RBL), *N*-way partial least-squares/RBL (*N*-PLS/RBL) and unfolded principal component analysis/RBL (U-PCA/RBL). The third is built on the alternating least squares and bilinear decomposition of an augmented matrix, known as multivariate curve resolution coupled to alternating least squares (MCR-ALS), which allows for the multi-linearity deviation of three-way data. Additionally,”. We think that these sentences can help readers better understanding of the paper. Please see **the red words in lines 14-41 in page 5.**

(4) Why does the retention region of five PAHs been divided into four sub-segments, but not three or two (Page 11 lines 17-20)?

Authors' Response:

Thank you. According to our experience, the retention region of five PAHs must be divided into at least three sub-segments. The chromatographic peak eluted sequence of five targeted PAHs was FLO, PYR, BaA, CHR and BbF in the eluted time range of 5.50 to 7.72 min. BaA and CHR seriously overlapped with each other, so they must be dealt with separately. Then, the analyte' eluted peaks before them were FLO and PYR, which can be dealt with together. The analyte' eluted peak after them was BbF, which must be dealt with separately. So, at least three sub-segments must be needed to analyze the obtained three-way data array. When we applied ATLD-MCR to deal with three sub-segments, good and reliable results could be obtained. However, when we applied MCR-ALS to analyze the same data, the first three-way sub-segment data could not provide good results (Other three-way sub-segments could provide good results). So, we further divided the first three-way sub-segment data into two sub-segment data for analysis of FLO and PYR, respectively. Then, satisfactory and

reliable results could be provided by MCR-ALS. Therefore, to maintain consistency, we dealt with the big three-way data array with four sub-segment data arrays using ATLD-MCR and MCR-ALS.

(5) This manuscript should be submitted after careful inspection, such as duplicate line numbers.

Authors' Response:

Here we are to you apologize for the confusion caused to you. In the manuscript, we inserted continuous line numbers as required, and the leftmost line numbers were automatically generated by the system. So, in the revised manuscript, we detected the inserted continuous line numbers.

(6) Although this manuscript was well-written, there are still some grammatical errors in the manuscript. For example, in page 16 lines 40-41, "...is taken into account" should be "...was taken into account". Therefore, language of this manuscript should be further polished.

Authors' Response:

Thank you for your careful reading and valuable suggestion. According to your suggestion, we have checked the whole manuscript carefully and corrected some grammatical mistakes. In page 15 lines 56-57, "...is taken into account" has been changed to "...was taken into account". In addition, we have also invited several colleagues who are skilled authors and reviewers of English language papers to revise the English of this paper. We hope that the language is now acceptable for publication.

(7) The format of references should be modified to meet the requirements of this journal.

Authors' Response:

Thank you for your careful reading. In the revised manuscript, we modified the format of references according to the requirements of Royal Society Open Science. Please see the References part of the revised manuscript.

Reviewer #2:

This manuscript described the application of gas chromatographic-mass spectral data to the analysis of polycyclic aromatic hydrocarbons in aerosol samples. Although the chemometric models are already known, the analytical application appears to be original, however the paper needs major revision.

Authors' Response:

We do appreciate your kind approval on our work. According to your valuable comments and suggestions, we have checked and revised the whole manuscript carefully. The responses to your comments and suggestions are summarized as follows point-by-point. We hope that you will be satisfied with our responses and modifications.

- Only three validation samples (Table 1) are not enough to gather reliable statistical parameters and conclusions based on t-statistics. You may need to significantly increase the number of validation samples. I suggest a minimum of 10 samples.

Authors' Response:

Thank you very much for your valuable comments and suggestions on our manuscript. The small number of validation samples was indeed the imperfect part of our experimental design. We were very much in agreement with your suggestion and also willing to do it. However, as you know, due to the influence of COVID-19, the difficulty of making experiment had increased in last year. In the study, the GC-MS experiment was done in National Ceramic Testing Center which located in Loudi City, Hunan province, China, far away from our city. Therefore, it is very difficult to make the GC-MS experiment there, please understand this practical difficulty. Besides, an apparent conclusion can be drawn from "Results and discussion" is that three samples of validation samples are representative, and all the T-test values for ATLD-MCR and MCR-ALS were reliable, which were lower than the reference values of 4.30 in the revised manuscript. Therefore, we ask that you can understand such a difficulty and we will take your valuable suggestions in our further research. Thank you once again!

- A GC-MS external reference method is mentioned several times in the text, but no details are provided for this technique. A comparison of the reference method and the one employed for three-way analysis is required, in terms of time, cost, equipment,

etc.

Authors' Response:

Thank you very much for your good suggestion, which will help us to improve the quality of our submission. So, more details about GC-MS external standard method were added into the revised manuscript (Please see the red words in lines 16-25 in page 8, lines 56-60 in page 12 and lines 3-7 in page 13), and a comparison between the reference method and the three-way calibration method such as ATLD-MCR, was provided in the revised manuscript. Please see the red words in lines 27-46 in page 14, lines 37-41 in page 15 and lines 45-49 in page 15.

- I do not really see the point in comparing ATLD, ATLD-MCR and MCR-ALS. ATLD is a trilinear decomposition technique which is known to be unreliable for non-trilinear data such as those presently studied. Ultimately, the analytical results from ATLD-MCR do not seem to be significantly better than those provided by MCR-ALS, which is the standard method for second-order chromatographic analysis of complex samples. I suggest removal of ATLD and ATLD-MCR from the text.

Authors' Response:

Thank you. We do agree with the point that ATLD is a trilinear decomposition technique which is known to be unreliable for non-trilinear data. So, in the revised manuscript, we detected the content of ATLD including the corresponding discussion on the theoretical and practical aspects of ATLD.

Our research purpose in the paper was to investigate the performance of ATLD-MCR for the analysis of second-order GC-MS data, which has not been studied by anybody until now. ATLD-MCR was a new algorithm which had been applied to deal with second-order liquid chromatographic data even in the presence of retention time shifts. However, no study so far has demonstrated that ATLD-MCR can process complex GC-MS data with severe baseline drifts, overlapped peaks and retention time shifts. Thus, ATLD-MCR was applied to analyze complex GC-MS data in the present work for the first time. The results of validation samples proved that ATLD-MCR could deal with complex second-order GC-MS data and provide satisfactory results. ATLD-MCR was also used to analyze actual aerosol samples for

the first time. To further confirm the reliability of the ATLD-MCR results, MCR-ALS was applied to analyze the same data and was compared with ATLD-MCR. It was found that ATLD-MCR could provide good results as the same as MCR-ALS. Additionally, ATLD-MCR fully utilized the advantages of ATLD and MCR-ALS. ATLD-MCR had advantages of directly decomposing three-way data array, the ability of dealing with non-trilinear data, providing pure chromatographic, spectral and relative concentration profiles of each chemical constituent of interest, second-order advantage and so on. In a word, ATLD-MCR was a great choice to deal with complex GC-MS data for the simultaneous determination of targeted PAHs in aerosols.

MCR-ALS is classical and widely used method in many fields, such as food analysis, environmental monitoring, pharmaceutical analysis and so on. MCR-ALS had been applied to resolve GC-MS data for the analysis of the volatile chemical constituents in Iranian *Citrus aurantium* L. Peel (F. Azimi and M. H. Fatemi, RSC Advances, 2016,6(112):111197-111209). MCR-ALS was also used to apportion the source of submicron organic aerosol at an urban background and a road site in Barcelona (Spain) combined with GC-MS (M. Alier, B. L. van Drooge, M. Dall'osto, X. Querol, Atmospheric Chemistry and Physics, 2013, 13(4):11167-11211). However, nobody reported that MCR-ALS combined with GC-MS was applied to quantify PAHs in aerosol samples. Therefore, MCR-ALS was also used to analyze the contents of PAHs in aerosol samples and was compared with the new algorithm of ATLD-MCR.

We think MCR-ALS and ATLD-MCR were comparable in dealing with non-trilinear three-way data because they were three-way calibration methods, and the GC-MS-based external standard method was only used as a reference method. The results of MCR-ALS could better verify those of ATLD-MCR, which could make the content of the paper more substantial and reliable. So, in the revised manuscript, we used ATLD-MCR to analyze the three-way GC-MS data and compared it with MCR-ALS.

- In the real samples, Table 3, most of the analytes are undetected. Moreover, those that are detected appear to show concentrations close to the limit of detection (which

in any case is an approximation, because the effects of slope and intercept have not been taken into account in estimating the LOD), and certainly below the limit of quantitation, which is not reported here. I guess most (if not all) concentration values quoted in Table 3 should be reported as “detected but not quantitated” because they are between the LOD and the LOQ. If this is the case, Table 3 is not required, and you can quote the results directly in the text. In a way, it is rather disappointing that no aerosol sample has been collected containing measurable amounts of the studied PAHs.

Authors' Response:

Thanks very much for your suggestion. According to your suggestion, we detected original Table 3. In the revised manuscript, we directly quoted the results of PYR predicted by ATLD-MCR, MCR-ALS and the GC-MS external standard method, and BbF predicted by MCR-ALS. Please see the red word in page 12 lines 56-60, page 13 lines 3-7 and 50-60 and page 14 lines 3-18.

- In Figures 3, 4 and 5, some of the ATLD and ATLD-MCR profiles show negative elements, which is not acceptable for signals and concentrations which are all non-negative. Only MCR gives physically interpretable results, derived from the wise application of the non-negativity constraint. One more reason to remove the ATLD and ATLD-MCR results and the corresponding discussion on the theoretical and practical aspects of these two latter algorithms.

Authors' Response:

Thank you. According to your suggestion, in the revised manuscript, we detected the content of ATLD including the corresponding discussion on the theoretical and practical aspects of ATLD.

However, the non-negativity constraint can also be applied on ATLD-MCR. So, we reprocessed the GC-MS data using ATLD-MCR with a non-negativity constraint. This time the non-negativity constraint was applied on the elution time profile when using ATLD-MCR. We also obtained physically interpretable results. The quantitative and statistical results newly obtained from ATLD-MCR were listed in Table 2 (rows 3-10) and Table 3 (rows 2-5). The chromatographic, spectral and relative concentration

profiles retrieved by ATLD-MCR were redrawn and show in Figures 3(A1, B1, C1), 4(A1, B1, C1) and 5(A1, B1, C1).

- The manuscript needs professional edition of the English.

Authors' Response:

Thanks very much for your suggestion. We have made careful improvements in the revised manuscript and tried to avoid any grammar or syntax errors. In addition, we have asked several other colleagues who are skilled authors of English language papers to check the English. We hope that the language is now acceptable for the review process.

Sincerely yours,

Ling Xu and Xiao-Hua Zhang